# A portable, ultra-low cost, open-source, pedal-controlled microinjector for laboratory use

**Victor H. Dominguez**[1,2,3], **Maxwell Frankfurter**[4,5], **Kevin B. Hayes**[4,6], **Mark L. Kahn**[4], **Martin H. Dominguez**[7,8,9‡]*

**1** Tulane University School of Medicine, New Orleans, Louisiana, United States of America, **2** Sam Houston State University Department of Family Medicine, Conroe, Texas, United States of America, **3** OakBend Medical Center, Richmond, Texas, United States of America, **4** Cardiovascular Institute, University of Pennsylvania, Philadelphia, Pennsylvania, United States of America, **5** Perelman School of Medicine at the University of Pennsylvania, Philadelphia, Pennsylvania, United States of America, **6** Rutgers Robert Wood Johnson Medical School, Rutgers University, New Brunswick, New Jersey, United States of America, **7** School of Biomedical Engineering, Science and Health Systems, Drexel University, Philadelphia, Pennsylvania, United States of America, **8** Department of Medicine, Drexel University College of Medicine, Philadelphia, Pennsylvania, United States of America, **9** Department of Cardiology, Tower Health, Phoenixville, Pennsylvania, United States of America

‡ Lead contact
* mhd47@drexel.edu

## Abstract

*In vivo* transduction or electroporation often requires sub-microliter volume injection of virus or recombinant DNA/RNA to a precise anatomic site. Two-hand manual manipulation of the injection needle and target tissues is dramatically faster than mechanical manipulation, but technically challenging for operators. Here, we present an all open-source, footswitch-actuated injector for nano- or microliter volumes. Our microinjector is simple, can be assembled in less than 2 hours with ordinary tools, does not require custom fabrication or soldering, and can cost less than $130 USD. This device is completely self-contained, pressure controllable, and offers both aspirate and discharge modes to facilitate multiple injections during the same procedure. Pulse-triggered discharges can reliably deliver nanoliter volumes with dispense errors of around 15%. By reducing technical and financial barriers, we anticipate this microinjector may inspire adoption of *in vivo* electroporation or genome editing across broader scientific communities where access may be presently limited.

## Introduction

Many experiments in developmental biology, neuroscience, and genetic engineering require delivery of very small volumes (often in the sub-microliter to nanoliter range) of reagents—viral vectors, plasmid DNA or RNA, morphogens, dyes—into specific anatomical locations *in vivo*. "Microinjection" has been essential to many biological techniques including creation of transgenic organisms across nearly all species, assisted reproductive technologies such as intracytoplasmic sperm injection (ICSI),

**Data availability statement:** All relevant data are within the paper and its Supporting Information files.

**Funding:** This work was supported by Sam Houston State University and OakBend Medical Center (to VHD); the National Institutes of Health Medical Scientist Training Program at the Perelman School of Medicine at the University of Pennsylvania and NIH (F30HL175934 to MF);NIH grant (R01HL153224 to MLK) and support from the Leducq Foundation (to MLK); NIH training grants (T32HL007731 and T32HL007843 to MHD); NIH grant (3R01HL164929-02S1 to MLK); NIH grant (K08HL171841 to MHD); and the American Heart Association (24TPA1285467 to MHD). The funders had no role in study design, data collection and analysis, decision to publish, or preparation of the manuscript.

**Competing interests:** The authors have declared that no competing interests exist.

manipulation of cells and organs for lineage tracing, and functional studies of cell physiology [1,2].

Over two decades ago, *in utero* electroporation (IUE) became a standard tool for probing gene function in the developing mammalian nervous system, enabling spatially and temporally controlled genetic manipulation in embryos [3,4]. IUE involves surgically exposing embryos in a pregnant rodent, microinjecting DNA/plasmid into the ventricular zone or other target region, then applying electric pulses to introduce the DNA into progenitor cells. Massive extension of *in vivo* electroporation to fields such as muscle biology, oncology, and reproduction has permitted new techniques and pioneering discoveries [5–8].

A variety of technologies have been developed to deliver small volumes of fluids in biomedical contexts, including needle-based injection systems, needle-free jet injectors, and other transdermal delivery platforms [9,10]. As a subset of these, microinjection applications—such as embryo manipulation, *in vivo* electroporation, and localized reagent delivery—typically require compact instrumentation capable of precise nanoliter-scale injections under manual microscope control. Precision in volume and anatomical targeting is critical: overshooting volume can damage tissue or produce imprecise spread, while too little may fail to transduce enough cells. Moreover, speed and ease of manipulation matter, especially when working in delicate contexts (embryos, small animals, brain), where minimizing exposure time, reducing animal stress, and maintaining sterility are crucial.

Existing tools for microinjection in research settings include both commercial and open-source devices. Several recent open hardware projects have made significant strides (Table 1). While these represent excellent progress, they often have limitations: some require custom fabrication (e.g., 3D printing), precise machining, or soldering; some depend on compressed or regulated pressurized air sources; some offer a single mode (dispense only) and lack an aspirate/fill operation. Most require two-handed operation or mouth pipetting, which can slow or complicate procedures when precise rapid injections are needed.

Here we present an all open-source, footswitch-actuated microinjector that addresses most of these limitations. The device is fully self-contained and does not rely on external compressed air supplies, yet provides controllable pressure for consistent operation. It supports fill/aspirate and discharge modes, enabling rapid sequential loading and dispensing of multiple reagents through the same needle during a single experiment, without requiring syringe exchange or mouth pipetting. Footswitch actuation frees both hands for precise handling of tissues and needles, increasing dexterity and eliminating unsafe practices such as mouth pipetting. Assembly requires less than two hours with only standard tools—no custom machining, 3D printing, or soldering—making it broadly accessible at a cost of approximately USD $130 (at time of writing). Despite this simplicity, the injector reliably dispenses volumes as low as ~25 nanoliters with errors of roughly 15%, offering a practical balance of sensitivity and accuracy for routine laboratory use.

By lowering both technical and financial barriers, this device aims to broaden access for researchers who need precise *in vivo* injections—for example

**Table 1. Open-source microinjection devices described in the literature.**

| Device | Dispense Volume | Actuation | Aspirate Function | Assembly Complexity | Cost | Notes/ Unique Features |
|---|---|---|---|---|---|---|
| Open-Source Microinjector [this work] | 25 nL – µL ± 15% | Footswitch, module provided for timed pulse generation | Yes (fill/ aspirate) | <2 h total, no soldering, no custom fabrication | <$130 | Puff injection for broad applications; multiple serial injections; validated small-volume delivery. |
| Openspritzer [11] | 10 pL – µL (pressure puff) | Arduino-controlled solenoid, button trigger | No | 3D-printed + electronics; soldering required | ~£360 | Puff ejection for neurons/ tissue; similar to commercial picoinjectors. |
| Arduino Intra-Cerebral Microinjector [12] | 0.5 µL – 1 mL | Arduino stepper-controlled syringe | No | Requires electronics, 3D printing; moderate build | <$200 | Neuroscience stereotaxic use; fine control over injection rate. |
| Zebrafish Embryo Injector [13] | ~nL scale, imprecise | Manual | No | Simple assembly; no calibration possible | Very low | Traditional mouth pipettors; teaching-friendly. |
| C. elegans Microfluidic Injector [14] | ~nL (worm injections) | PDMS microfluidics with passive positioning | No | Cleanroom fabrication; specialized | Moderate | High-throughput transgenesis; organism-specific. |
| Drosophila Embryo Microinjector [15] | 30 pL ± 10 pL | Pressure pulses into PDMS chip | No | Microfabrication required | Moderate | Localized injection in fly embryos. |
| Open-Source Syringe Pumps [16] | nL – mL (syringe-dependent) | Stepper-driven syringe, software control | Yes (full syringe cycle) | 3D printing, Arduino/ RPi, soldering | ~$200 | Versatile; general liquid handling. |
| 3D-Printed Autoinjector [17] | 0.3–1 mL | Spring-driven | No | 12 h print/assembly; moderate effort | ~$7 materials | Meets ISO standards; designed for insulin self-injection. |

Values represent approximate ranges as reported in the cited publications.

electroporation, viral transduction, lineage tracing, optogenetics, or genome editing—especially in settings without high budgets or specialized fabrication tools.

## Materials and methods

### Microinjector assembly

A comprehensive list of parts and tools required for complete assembly, including annotations and alternatives, is listed in S1 Table. Step-by-step instructions for assembly are provided in the Results section and figures. In brief, the enclosure was first drilled with the appropriate holes for switches and connectors, all wire tips were stripped, the tubing was connected between the pump, solenoids, and panel-mounted hose barb, and wires were connected.

### Needle holder assembly and operation

Glass microinjection needles were pulled from single barrel borosilicate glass capillaries (1B120-4 and 1B100-3, World Precision Instruments) on a PN-3 Electrode Puller (Narishige) utilizing a platinum–iridium filament with a round cross-section hole. Three needle holder assemblies are recommended for use with our microinjector (Fig. 2B): World Precision Instruments (WPI) 5430-ALL, Tritech Research MINJ-4, or our custom home-built needle holder.

1. For WPI 5430-ALL, connect the included hose to the 2 mm output barb of the microinjector. The perforated metal cover bolt is turned to tighten or loosen the grip on the inserted glass capillary, allowing secure mounting or rapid exchange. WPI provides instructions for using different capillary sizes with 5430-ALL.

2. For Tritech MINJ-4, a hose adapter is required for connecting the PTFE tubing to the 2 mm barb of the injector. We suggest constructing an adapter from an 8–10 mm aluminum M3 spacer and two M3 barbs (3 mm and 2 mm). The 3 mm barb is threaded onto one end of the spacer and the 2 mm barb onto the other. A short segment of 1/16″ ID tubing

is connected to the larger barb, and 2 mm PTFE tubing to the smaller barb, forming a stable transition to the injector. Use a heat gun where necessary to soften tubing and ensure durable, airtight connections.

3. For our custom glass capillary microinjection needle holder, follow instructions in the Results section for assembly. For capillary loading, slightly unscrew to loosen the empty hose barb, insert glass capillary until it cannot be pushed further into the barb, and tighten barb to secure. To remove, slightly loosen capillary-attached hose barb to loosen the capillary, and remove. If trouble is encountered at capillary attachment, completely unscrew hose barb to remove rubber o-rings and capillary, and reassemble according to initial instructions (Fig 4G-K).

## Microinjector operation

For routine use, the completed injector is powered by a 5 V, 3 A USB-C adapter. A footswitch connected through the GX16 port provides momentary actuation of aspiration or injection. Pressure is adjusted using the front-panel potentiometer dial, which controls the pulse width modulation (PWM) duty cycle and therefore pump speed. Airflow direction is selected by a toggle switch: when aspirating, the solenoid valves connect the pump inflow to the needle; when injecting, the pump outflow is directed to the needle.

The injector operates by coupling a miniature diaphragm air pump with two three-way solenoid valves that control the direction of airflow through the tubing network. The diaphragm pump generates positive pressure at its outlet and negative pressure at its inlet during each pumping cycle – irrespective of motor polarity. Therefore, a front-panel toggle switch determines which side of the pump is connected to the microinjection needle by altering solenoid position, not pump motor polarity. In aspirate mode, the solenoid valves route the pump inlet to the needle, producing suction that draws liquid into the capillary. In inject mode (default operating mode with solenoids unpowered), the valves instead connect the pump outlet to the needle, producing positive pressure that expels fluid from the capillary tip. Pump activation is triggered by the external footswitch, which simultaneously powers the pump motor and switches the solenoid valves as required, allowing rapid transitions between aspiration and injection while both hands remain free for manipulation of the tissue and needle.

Example setup and operation of the injector and needle holder are shown in S1 Video. To begin an experiment, attach the desired needle holder to the 2 mm exterior hose barb of the microinjector, softening tubing ends with a heat gun if necessary to ensure airtight connections. Insert appropriate pulled glass capillary (i.e., microinjection needle) into the holder according to its specific instructions (see Results, Fig 4). To aspirate reagent, set the toggle switch to aspirate / fill (closed switch) and depress the footswitch until the desired volume is drawn. To inject, set the toggle switch to inject mode (open switch) and again depress the footswitch. Rapid switching between aspiration and injection is possible without removing the needle, enabling sequential loading and dispensing of different reagents.

Between sessions, tubing and capillaries should be flushed with distilled water and dried to prevent clogging or contamination. The device should be inspected for leaks, especially at barb connections, and tubing segments replaced as needed. To check for leaks, ensure that a needle or glass capillary is installed at the outlet, place device in inject mode and activate the footswitch (approximately 2 s) to pressurize the system, while observing for signs of leakage such as audible air escape, failure of the system to reach the expected pressure, or rapid pressure loss after the pump stops. If leakage is suspected, tubing should be re-seated firmly on the barbed fittings or replaced as needed.

## Pulse generator assembly and operation

An optional pulse controller was developed to provide reproducible, tightly-timed control of pump activation. The controller integrates an inexpensive ZK-PP1K pulse generator and HW-548 MOSFET switching module within a compact enclosure, powered alongside the microinjector through a USB-C connection. Timing signals from the pulse generator are gated to the pump motor via the MOSFET board, while the existing potentiometer dial on the injector sets operating pressure. Users can connect one or two footswitches to the controller: the first initiates timed pulses as programmed on

the ZK-PP1K, while the second allows manual bypass operation for aspiration mode or for freely-timed injections. Detailed wiring diagrams and assembly instructions are provided (see Discussion).

### Timing and precision considerations

Typical aspiration and injection events occur on millisecond-to-second timescales depending on pressure setting, needle geometry, and fluid viscosity. For low-viscosity aqueous solutions (≈1 cP), aspiration of several microliters into a glass capillary typically requires ~1–10 s at moderate pressure settings, while pulse-controlled injections of nanoliter droplets occur over 5–10 + ms depending on desired volume. The system is compatible with a wide range of laboratory injectates including aqueous buffers, viral suspensions, plasmid solutions, and low-viscosity oils. Highly viscous fluids require the highest pressure settings and longer pulse durations, resulting in larger minimum droplet volumes.

Prior to experimental use, users should verify airtight tubing connections and inspect the system under pressure for visible leaks at tubing junctions or barbed fittings. Because injection volume depends on needle geometry—particularly tip diameter—users are encouraged to perform simple calibration experiments for each needle type by measuring droplet volumes at several pulse durations and pressure settings. This calibration step allows reliable mapping between pulse duration and delivered volume under the specific experimental conditions.

### Quantification of injections into non-enclosed liquids

For injector performance testing, timed pulses were programmed on the controller to deliver 3–5 replicate injections at a given pulse duration. Durations ranged from 1 ms to several seconds, with each pulse separated by 3-second intervals. PWM settings "low" and "medium" were arbitrarily established ("high" was equivalent to maximum 100% duty cycle PWM) on the potentiometer dial, and static measurements of pressure were made with a commercial-grade digital low pressure gauge connected to the output tubing with a tee hose barb. For each PWM setting of the injector (low, medium, and high/maximum), pulse durations were incrementally increased to span the range of stable injection. Low-viscosity injectate was modeled using distilled water mixed with red cabbage extract, which represents a biodegradable, visually contrasting dye, with fine particulate matter remaining after dissolution in water. High-viscosity injectate was modeled with 100% natural maple syrup (dynamic viscosity ~142 cP at room temperature) [18,19], which on empirical testing could only be reproducibly injected at "high" (maximum) pressure (i.e., PWM 100% duty cycle) setting.

Static pressure measurements were initially made with needle attached, into "air" unloaded. All volume-quantified injections were dispensed into a bath of paraffin lamp oil (Ner Candles) to form spherical or near-spherical droplets. Static pressure measurements were again made when steady-state pressure had been reached during each injection of low-viscosity injectate into lamp oil. A macro-objective digital camera recorded each injection with a millimeter-ruled scale in frame. Droplet diameter and length/width were measured, and droplet volumes were estimated by assuming approximate sphericity. Only stable injection settings were subjected to serial dispense and quantify procedures. Estimated volumes were plotted with matplotlib in python (source data and scripts available in S1 File – Data and Code).

### Microinjector troubleshooting

This section provides practical notes on common issues encountered during assembly or operation of the microinjector and needle holders.

- Residual aspiration after switch release: Decrease aspiration pressure and fill capillaries more slowly.

- Inspect tubing junctions under pressure for visible droplets or pressure loss. Re-attach tubing firmly to barbs (softening tubing with a heat gun if necessary) and replace tubing if repeated leakage occurs. A small amount of dilute soapy water can be applied to suspected junctions to visualize escaping air bubbles during pressurization via footswitch activation.

- Capillary not seating securely in holder: Adjust grub screw depth (custom holder) or re-seat O-rings. For commercial holders, confirm correct capillary size.

- Inconsistent aspiration/injection volumes: Flush capillaries with distilled water to remove debris or residue. Use maximum pressure injection to forcefully eject all remaining fluid and empty capillary. Inspect for partial clogs at the tip.

- Reduced pump performance: Remove needle holder at panel barb of microinjecter, then use finger to confirm strong aspiration and injection pressure when pressure knob is at max position (clockwise).

## Results

### Design and fabrication objectives of a portable microinjector

To overcome the limitations of existing microinjection systems, we designed a portable microinjector that combines simplicity, reliability, and accuracy. Our objectives were fourfold: 1. Accessibility — assembly in less than two hours using only off-the-shelf components and standard hand tools, with no soldering, 3D printing, or machining. 2. Self-containment — elimination of external compressed gas sources by using a miniature diaphragm pump powered by a standard 5 V DC supply. 3. Dual-mode operation — support for both aspiration and injection modes, allowing the user to rapidly switch between filling and dispensing without manual syringe handling. 4. Hands-free actuation — external triggering through a footswitch, freeing both hands for precise manipulation of tissues and needles.

The resulting design is shown schematically in Fig 1. A miniature diaphragm pump is driven by a NE555-based pulse width modulation (PWM) controller and N-channel MOSFET, allowing user-controlled modulation of aspiration and injection pressure via the potentiometer knob (Fig 1A). Airflow direction is determined by a pair of three-way solenoid valves that switch between aspirate and inject configurations (Fig 1B–C). Empirical testing revealed that high-side switching of the solenoids with the footswitch (to switch to the aspiration airflow pattern) together with pump activation was essential to prevent unintended residual vacuum in the tubing even after footswitch release (previous beta-test versions had continuously on or off solenoids that did not open/close with the footswitch).

The completed injector (Fig 2A) is housed in a 115 × 90 × 55 mm enclosure, with a USB-C power port, GX16 footswitch connector, and 2 mm hose barb output. Integrated controls include a potentiometer knob for adjusting pressure and a toggle switch for selecting airflow direction. The device can be used with several needle holder options—commercial WPI or Tritech holders, or a custom home-built design (Fig 2B)—providing flexibility across different glass capillaries and experimental setups.

### Microinjector assembly

Assembly of the injector is straightforward and requires only basic tools (Fig 3B). The enclosure is first prepared by marking, pre-drilling, and drilling holes for the connectors and switches (Fig 3C–E). Wires are then cut, stripped, and bundled for later connections (Fig 3F). Panel-mounted components, including the toggle switch, USB-C power port, and GX16 footswitch socket, are installed in their respective positions (Fig 3G).

Tubing is cut to defined lengths (Fig 3H) and joined at a polypropylene three-way barb fitting (Fig 3I), which is connected to the paired solenoid valves and pump (Fig 3J–L) to establish bidirectional flow depending on the valve position (pay careful attention to how tubing is connected to solenoid valves). The assembled pump–valve–tubing unit is positioned inside the enclosure and attached to the 2 mm panel-mounted hose barb (Fig 3M–N). Durable connections are achieved by briefly softening tubing ends with a heat gun before advancing further onto the barbs, which improves sealing and injector reproducibility.

Electrical connections are made without soldering (Fig 1D), using twist-on caps to join pump and solenoid leads with the footswitch, toggle switch, and USB-C power input (Fig 3P–X). The PWM motor controller is mounted to the lid,

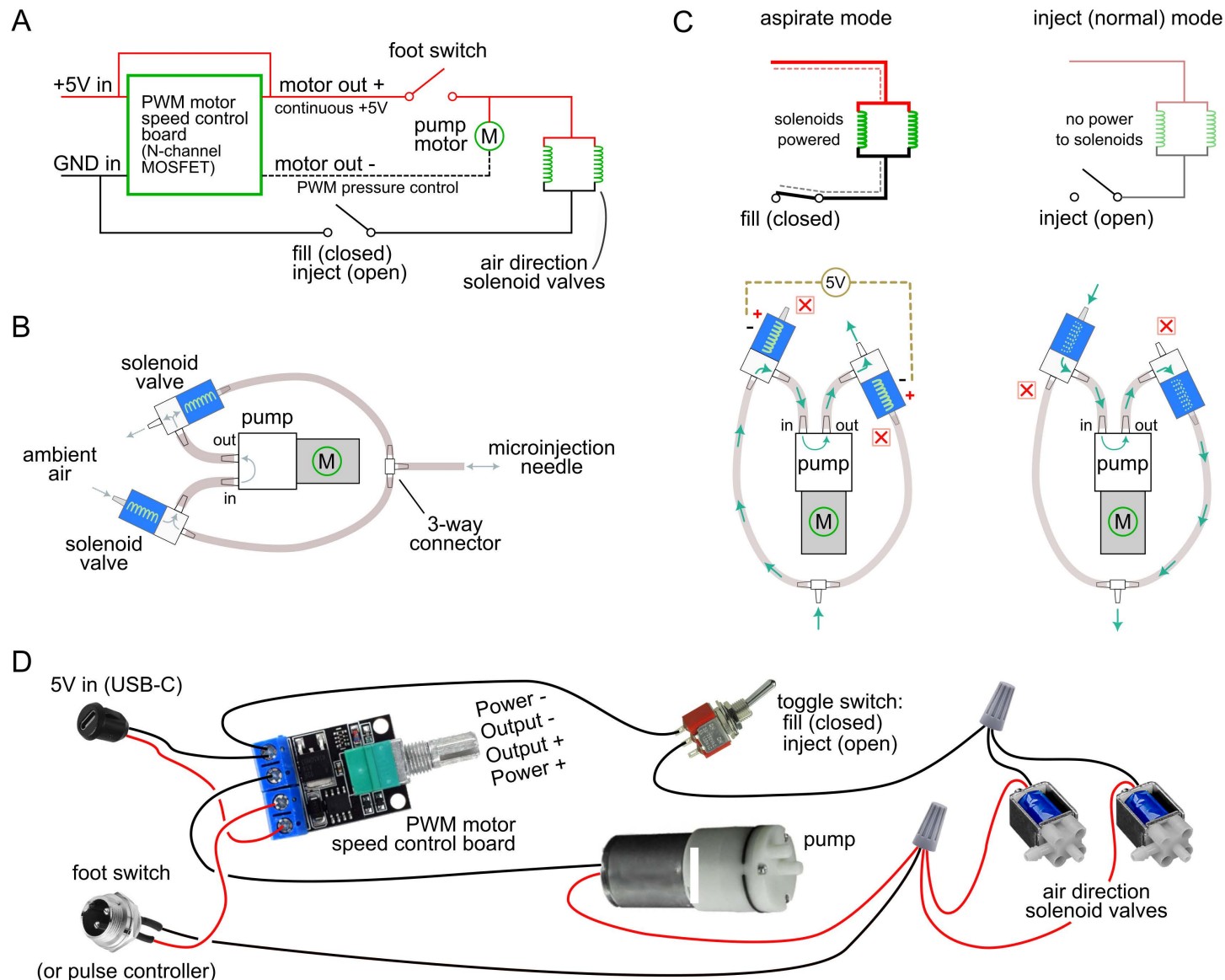

**Fig 1. Basic design and function of a portable self-contained, externally-actuated microinjector. A**. Overall electronics block diagram of the injector module. The device is powered by a 5V, 3A external AC-DC switching power supply, which supplies all control, pressurization, and airflow. Motor speed and therefore pump pressure is controlled by a simple, off-the-shelf PWM Module utilizing a NE555 timer to generate a dial-controllable pulse train feeding a P2003BDG or LR7843 N-channel MOSFET. The result is high-frequency, low-side PWM of the pump motor, with high-side motor control by a foot pedal or other external switch. Direction control of airflow (aspirate or inject mode) is managed by a pair of solenoid valves that are separately switched, on for aspirate, off for dispense. Note the solenoids themselves are high-side switched on/off with the footswitch as well, as empiric testing found this configuration was necessary to release negative pressure built up in the tubing (when needle filling is completed), which would otherwise be transmitted to the injection needle (creating continued aspiration) even after the high-side switch was released. **B**. Tubing schematic showing the unidirectional mini air pump, which is connected to two three-way mini solenoid valves that control airflow direction to the microinjection needle. **C**. In aspirate mode, solenoids are actuated, closing the bottom metal ports and opening the top plastic ports, a configuration that results in the pump inflow port being connected to the microinjection needle. In inject/ dispense mode, solenoids are not powered, closing the top plastic ports of the valves and opening the bottom metal ports, which causes the pump outflow port to be connected to the microinjection needle. **D**. Wiring diagram of the main microinjector, for user assembly. Detailed parts list is available in S1 Table.

## A

main injector

optional pulse controller

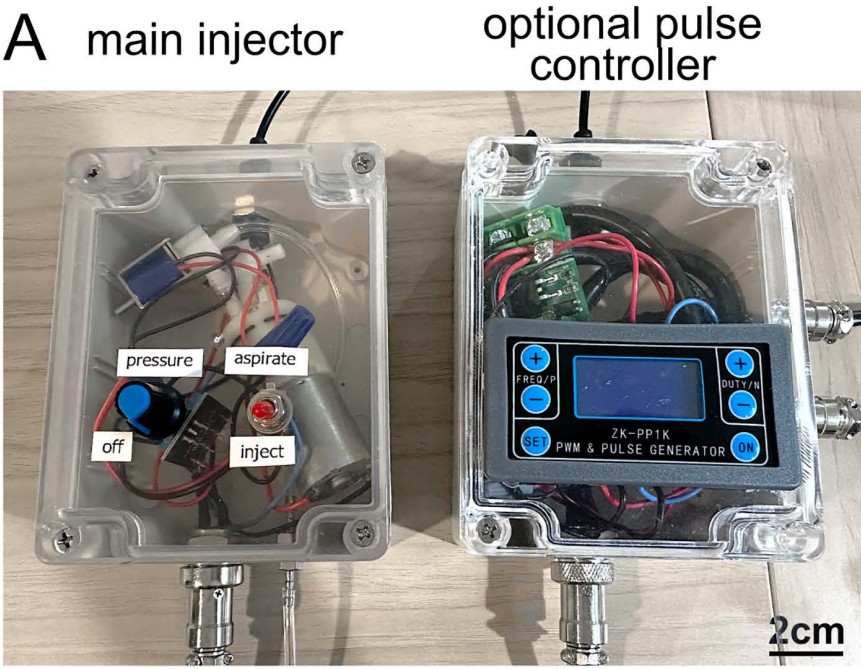

## B

needle holder options

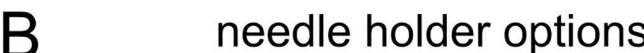

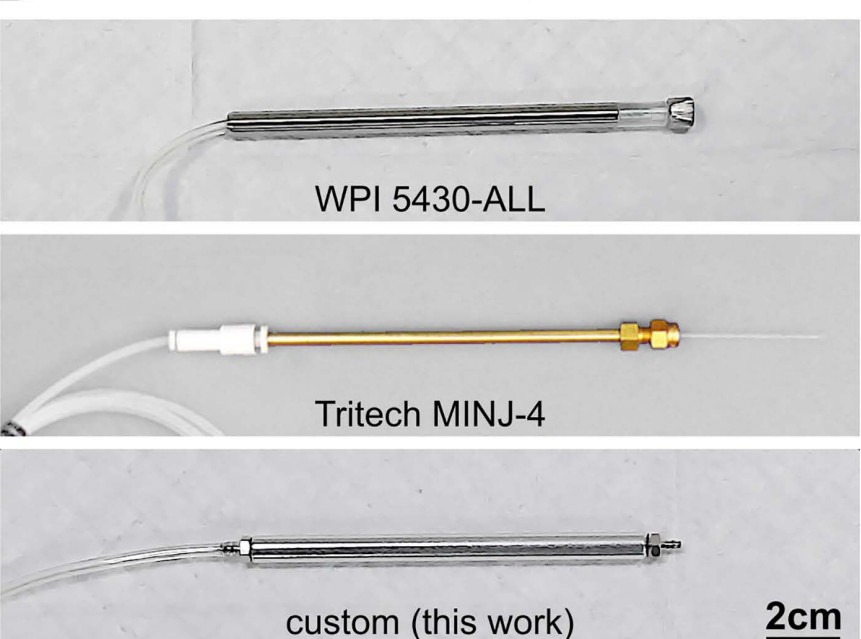

**Fig 2. Completed microinjector and needle holder options. A**. The completed microinjector fits ($85, assembly described below) into a 115 x 90 x 55 mm project box, with a USB-C power connection, a 2-pin GX16 aviator port for pump control, and a 2 mm hose barb fitting for pressure output to microinjection needle. Built-in controls include a injection/aspiration potentiometer knob, which controls motor PWM (aspirate/inject pressure), as well as a SPST toggle switch that controls solenoid valves for airflow direction (aspirate or inject). Optional pulse control module ($75, requires additional assembly, see Fig 7 and 8) also fits into a second similar package, and interfaces with the main injector via its footswitch GX16 port. **B**. Needle holder options include Word Precision Instruments 5430-ALL ($165 part), Tritech Research MINJ-4 ($78 total, not tested here), or custom home-built needle holder ($36 total, assembly described below).

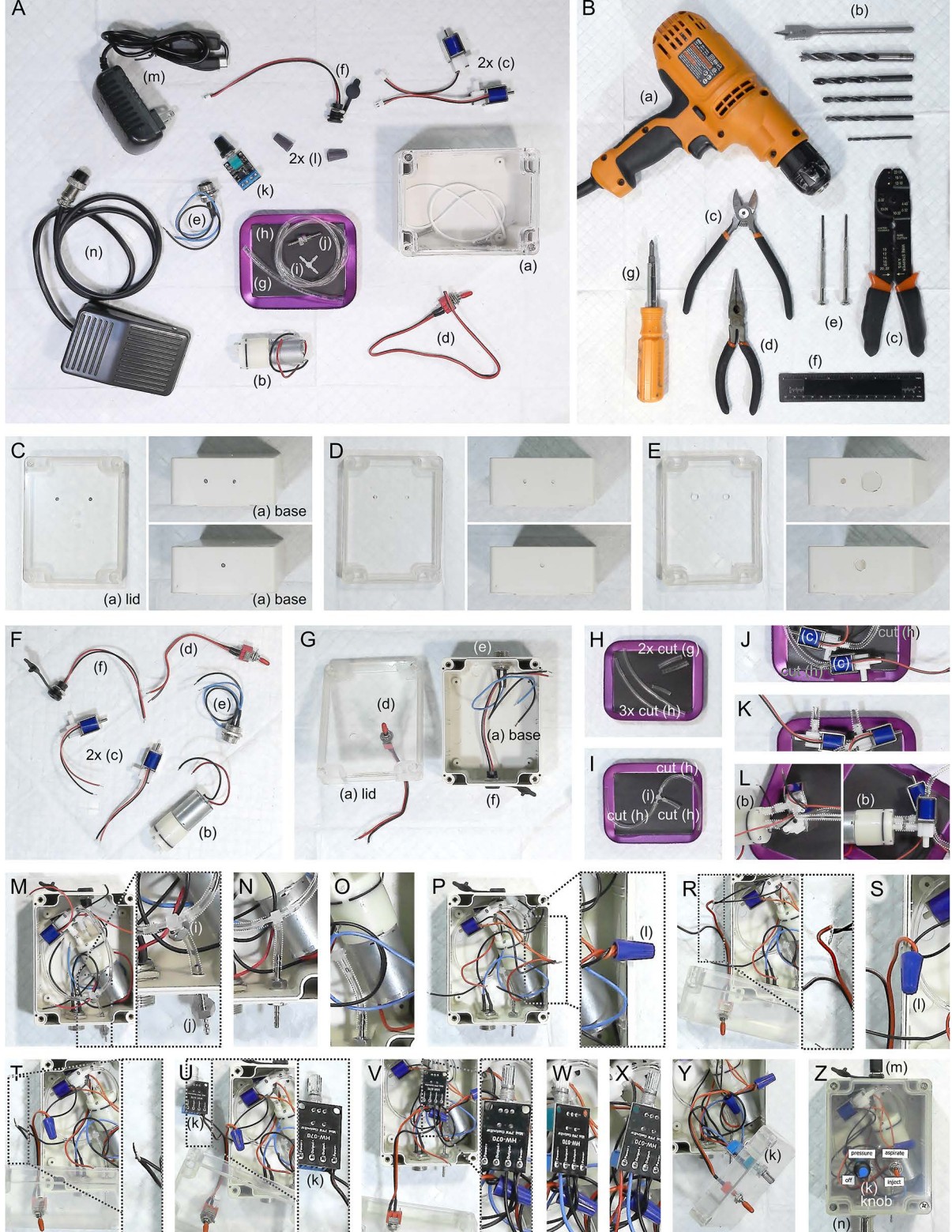

**Fig 3. Assembly of the microinjector. A**. All parts of the main injector are laid out before assembly: (a) Waterproof plastic junction box 115 x 90 x 55 mm; (b) 3.7–6 V Mini diaphragm air pump; (c) DC 5V 3-Way Mini Solenoid Air Valve; (d) Mini SPST toggle switch with Pre-soldered Wires; (e) GX16 2-Pin Male Aviation Socket Connector with Wire; (f) USB-C Female Charging Jack Port; (g) 10 cm of 1/8" ID x 3/16" OD x 1/32" Wall Ester-Based

Tubing; (h) 30 cm of 1/16" ID x 1/8" OD x 1/32" Wall Ether-Based Tubing; (i) 1/16" x 1/16" x 1/16" Natural Polypropylene Tee Hose Barb 3-way Fitting; (j) 2 mm Hose Barb Bulkhead Stainless Pipe Fitting; (k) Mini PWM 4.5-35V DC Motor Speed Controller Module; (l) Twist-on quick wire connector cap; (m) USB-C 5V 3A [Raspberry Pi 4] Power Adapter; (n) Foot Pedal Switch Nonslip Momentary 2M Wire 2Pin GX16 Female Connector. **B**. Majority of tools required for main injector assembly: (a) Variable speed drill; (b) drill bits (top to bottom): 5/8" spade (for e), 3/8" (for f), 9/32" (for k), 15/64" (for d), 7/32" (for j), 1/8" (for pre-drilling); (c) diagonal pliers or [preferably] wire stripping tool; (d) long-nose pliers; (e) miniature screw drivers; (f) ruler; (g) Phillips head screwdriver. Optional not-shown heat gun may help soften tubing for stronger connections during assembly. **C-E**. Beginning of assembly. Mark (**C**), pre-drill (**D**), and drill (**E**) all holes in the enclosure base (7/32" and 5/8" spade holes on same end, 3/8" regular hole on opposite end) and lid (15/64" and 9/32"). Note in **E**, lid is inverted. **F**. Cut and strip all wires. **G**. Install toggle switch into lid, USB-C power port and GX16 plug into enclosure base. **H**. Cut two segments of 1/16" ID tubing to 4" (10 cm) each, one segment of 1/16" tubing to 1" (2.5 cm), and two segments of 1/8" ID tubing to 1" (2.5 cm) each. **I**. Connect three segments of 1/16" ID tubing to 1/16" 3-way Tee fitting, with 1" segment to the middle port. Use heat gun if available. **J**. Connect distant ends of 1/16" ID tubing to opposite end ports (metal and plastic) of the two solenoid valves. If needed, sand plastic port barbs to allow hose to fit over. **K**. Connect two segments of 1/8" ID tubing to each side port of the two solenoid valves. **L**. Connect outflow (middle) port of pump to solenoid with open plastic top port, and connect inflow (outboard) port to solenoid with open metal bottom port. **M**. Bring pump, tubing, and valve assembly into enclosure base, and connect distal end of short 1/16" ID tube segment to the 2 mm panel mount hose barb. **N**. Screw to install 2 mm hose barb into enclosure. **O**. Transfer motor casing over the top of the connected 1/16" ID tubes. **P-Q**. Twist connect three motor wires (red here) to the right pin wire of the GX16 socket, and install twist-on cap. **R-S**. Twist connect two other solenoid motor wires (black here) to one wire of the toggle switch, and install twist-on cap. **T-U**. Twist connect other wire of toggle switch to the ground (black wire) from USB-C port, and insert twisted wire bundle into the Negative (-) Input Power (or GND in) screw terminal of the PWM motor controller board. Secure wires by screwing terminal into locked position. **V**. Insert other unconnected pump motor wire (black here) to the Negative Output (-) terminal of the PWM board, and screw into locked position. **W**. Insert wire from left pin of GX16 socket into the Positive Output (+) terminal of the PWM board, and screw into locked position. **X**. Insert the positive +5V (red) wire from the USB-C port into the Positive Input Power (or +V in) terminal of the PWM board, and screw into locked position. **Y**. Install PWM board onto enclosure lid, secured by its potentiometer knob shaft. **Z**. Close and secure lid to enclosure. Label ports, knob, and switch.

secured via the potentiometer knob shaft (Fig 3Y), and the enclosure is closed to complete the build (Fig 3Z). The process can be completed in two hours, yielding a fully functional injector at a total cost of USD $130.

## Needle holder assembly

To provide a flexible and low-cost alternative to commercial capillary holders, we developed a custom design that securely clamps glass microinjection capillaries of various diameters (Fig 4). The holder consists of a long aluminum M4 spacer body, with a recessed grub screw to support the capillary, and a threaded M4 barb on the proximal end to connect to the microinjector hose. A very short PTFE sleeve (~2 mm) is placed into the grub screw's Allen socket as the backing for the glass capillary. One or more size-matched O-rings are installed around the base of the capillary to ensure a tight seal, and to convert longitudinal forces from the screwed-in barb into radial compression to hold the capillary. The M4 threaded barb on the distal (capillary) end is tightened to compress the O-rings onto the capillary.

Assembly of the custom needle holder begins by inserting the PTFE insert into the grub screw and threading the grub screw into one end of the M4 spacer with thread-lock adhesive (Fig 4B–E). Glass capillaries are fitted with O-rings and coupled to drilled or un-drilled barb fittings matched to their outer diameter (1.0–2.0 mm; Fig 4F–I). The capillary–barb assembly is then threaded into the spacer. Grub screw depth is critical, and should be adjusted to achieve secure compression without overtightening (Fig 4J).

Capillaries can be repeatedly exchanged by loosening and re-tightening the barb fitting (Fig 4K), or alternatively, multiple holders can be prepared for different capillary sizes. The proximal end of the spacer connects to 1–2 m of 1/16″ ID tubing for attaching to the injector (Fig 4L). This modular approach allows stable and reproducible capillary mounting at a lower cost than commercial holders.

## Injector performance and reproducibility

Key steps of microinjector operation are detailed in Materials and Methods. Although the microinjector is typically actuated directly by the footswitch, this arrangement proved unsuitable for careful performance characterization because human timing variability dominated the output. To address this, we developed a compact, low-cost pulse controller that provides reproducible, millisecond-scale activation of the pump. The controller houses an inexpensive ZK-PP1K pulse generator

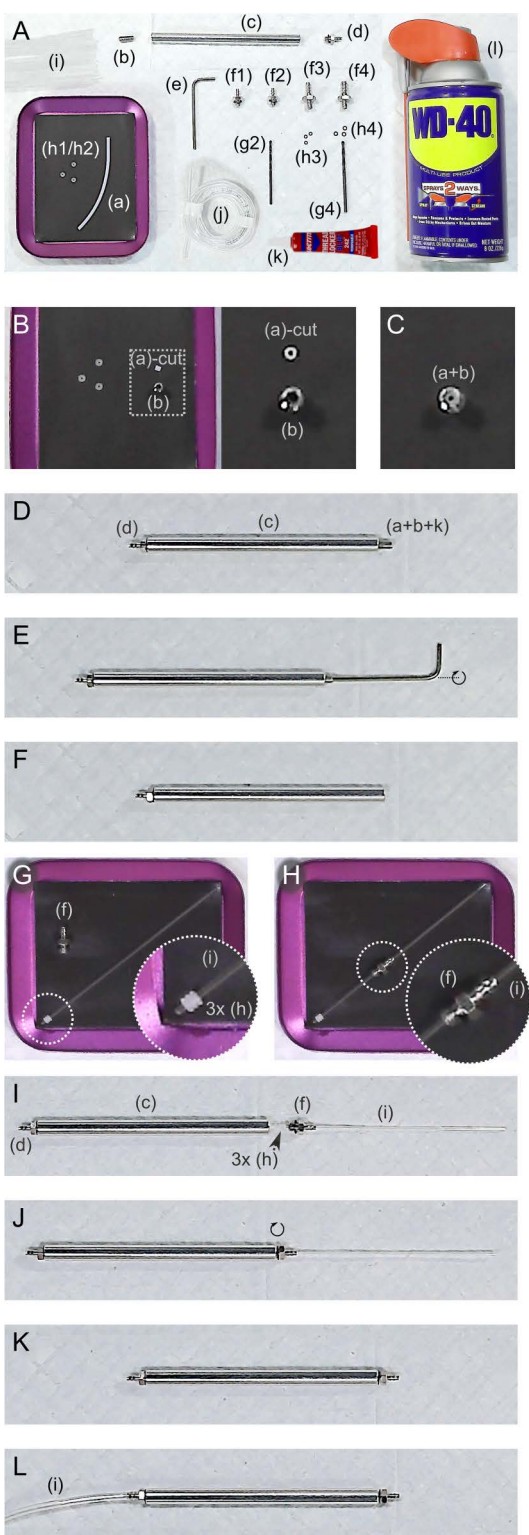

**Fig 4. Assembly of the custom home-built glass capillary needle holder. A**. All parts and additional tools required for the custom glass capillary holder: (a) 1 mm ID/ 2 mm OD PTFE tube; (b) 10 mm M4 through-hole air-out Allen socket grub/set screw; (c) 100 mm Aluminum spacer M4 round

Standoff rods; (d) Mini Barb Fitting "M-4AU-3(3-M4)"; (e) M4 Allen socket wrench; (f1) Mini Barb Fitting "M-4AU-3(3-M4)", undrilled, for 1.0 mm glass capillary; (f2) Mini Barb Fitting "M-4AU-3(3-M4)" drilled with 1/16" bit (g2) for 1.2 mm glass capillary; (f3) Mini Barb Fitting "3-M4", undrilled, for 1.5 mm glass capillary; (f4) Mini Barb Fitting "4-M4", drilled with 5/64" bit (g4) for 2.0 mm glass capillary; (h1/h2) Silicone O-ring CS 1 mm, OD 3 mm, for 1 or 1.2 mm glass capillary; (h3) Nitrile O-ring OD 2.5 x ID 1.5 x CS 0.5 mm, for 1.5 mm glass capillary; (h4) Nitrile O-ring OD 2.8 x ID 1.8 x CS 0.5 mm, for 2.0 mm glass capillary; (i) glass capillaries – OD 1.2 mm shown; (j) 1–2 m of 1/16" ID x 1/8" OD x 1/32" Wall Ether-Based Tubing; (k) Thread Locking adhesive for grub screw; (l) spray lubricant for drilling barbs for attaching 1.2 mm (f2/g2) and 2.0 mm (f4/g4) glass capillaries. **B**. Cut a small segment of PTFE tubing to 2 mm length. **C**. insert short segment of PTFE tubing into the open (Allen socket) end of the 10 mm M4 grub screw. **D**. Completely screw mini barb fitting "M-4AU-3(3-M4)" into one end of the aluminum 100 mm M4 spacer. Sparingly apply thread lock to the threads of the M4 grub screw, and begin to screw into other end of the 100 mm M4 spacer. **E-F**. Use Allen key to continue screwing grub screw into the aluminum M4 spacer; continue until grub screw is approximately 2 mm recessed into the M4 spacer. **G**. Place three of the matching-sized O-rings on one end of the glass capillary of user choice, and park them about 1 mm from the end of the capillary. **H-I**. Insert the matching-sized drilled or un-drilled hose barb onto the other end of the glass capillary, and bring the O-ring/capillary/barb assembly onto the open end (with recessed grub screw) of the M4 aluminum spacer. **J**. Screw the capillary-attached hose barb onto the end of the M4 spacer. It should tighten to stop just under 1mm from the end of the aluminum M4 spacer. If it tightens flush, remove and unscrew the grub screw less deep in the M4 spacer. If it cannot tighten securely, remove and screw in the grub screw deeper into the M4 spacer. Repeat until it tightens securely but is not completely flush with M4 spacer. **K**. Slightly unscrew to loosen the capillary-attached hose barb, and remove glass capillary. Whenever change in capillary size is needed, use techniques from **G-K**, or alternatively prepare several holders for each capillary size. **L**. Connect 1-2 m of 1/16" ID tubing to the opposite end of the needle holder. Connect other end of tubing to the microinjector.

and MOSFET switching board, powered in parallel with the injector. It allows the operator to program single or repeated injection pulses of defined duration, while the potentiometer dial on the injector maintains its function to set the pressure. Two footswitches can be connected: one for triggering programmed pulses and one for manual bypass operation (for aspiration or unscheduled injections). This design ensures that performance measurements reflect the true limits of the injector, rather than variability in operator reflexes.

Using the pulse controller, we systematically tested injection performance across fluids, pressures, and pulse durations. Glass needles were pulled from 1.2 mm outer diameter glass capillaries, and tips were shaped to 90–236 µm diameter without precision microforging (Fig 5A), suitable for a range of laboratory *in vivo* microinjection applications (but excluding pronuclear or oocyte injections). Because hydraulic resistance at the needle tip strongly influences flow rate, injection volume depends on the inner diameter of the needle tip; therefore, each needle geometry may require independent calibration of pulse duration versus delivered volume.

Experimental injections were delivered into a lamp oil bath and recorded with a macro camera for volume estimation (example pulse-controlled injections are shown in S2 Video). For low-viscosity injectate (~1 cP distilled water with cabbage extract), reproducible droplets were obtained across all pressures with carefully time-limited injections (Fig 5C-D). With different needles attached, static steady-state pressure measurements were made during injection (Fig 5B) with needle either unloaded into room air ("Air"), or with low-viscosity injectate into the bath ("Loaded") as would be quantified across settings. Interestingly, only PWM setting ("low," "medium," or "high") had a substantial effect on tubing pressure, as increasing needle diameters only lowered pressure at the top end of the range. "Loaded" status was associated with consistent but not marked decreases in observed pressure. These findings confirmed that the pump motor PWM setting was most suitable variable for controlling pressure at the needle. For clarity, subsequent analyses refer to low, medium, and high pressure settings that correspond to progressively higher PWM duty cycles on the pump controller (as selected through the motor control dial). The 112.5 µm diameter needle was selected for further testing.

The relationship between pulse duration and volume dispensed was monotonic: longer pulses yielded larger droplets, but the operational window shifted with pressure (Fig 5A). At high pressure (corresponding to the highest PWM duty cycle), very short pulses (≤10 ms) reliably produced nanoliter-scale droplets. At medium and low pressure, stable dispensed droplets required longer pulses, which paradoxically meant that the smallest achievable stable volume would increase as pressure decreased. For example, the smallest reproducible droplet at high pressure was ~22–25 nL, while at lower pressures the minimum stable volumes were larger (~45–50 nL).

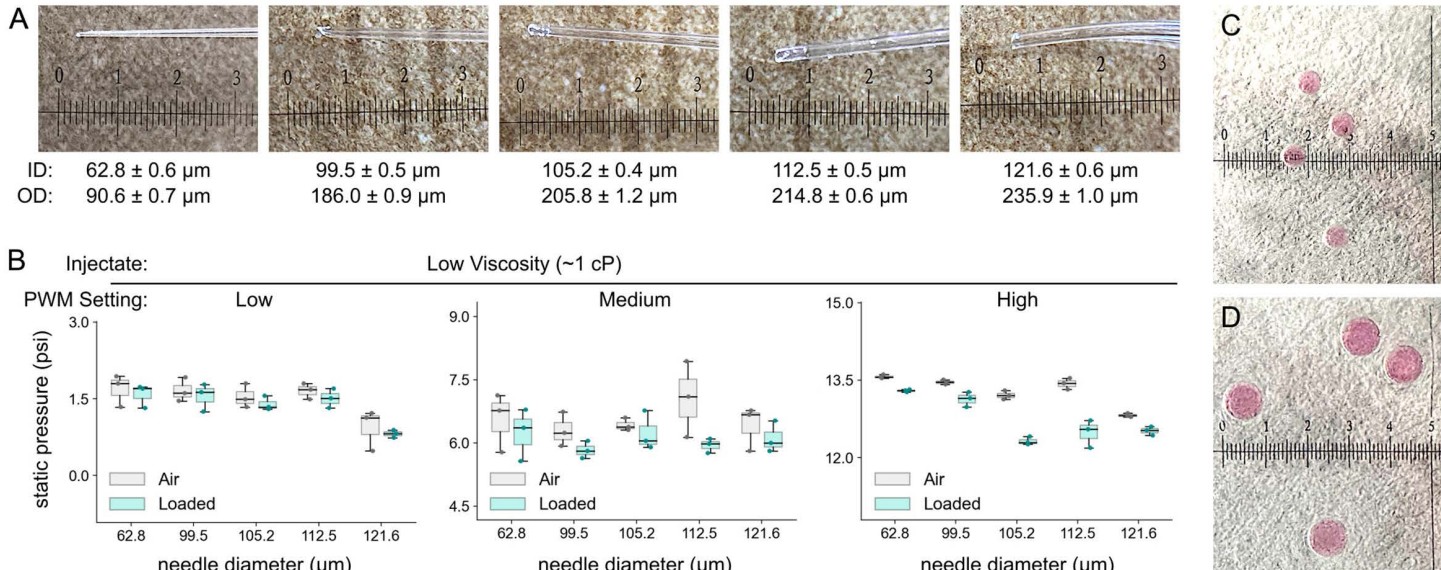

**Fig 5. Experimental setup for assessment of open-source microinjector performance. A**. 1.2 mm outer diameter borosilicate glass capillaries were pulled on a micropipette puller, and manually shaped into needles without precision microforging. **B**. Steady-state pressure measurements in the needle holder tubing were made with needle attached, and inject held on the PWM setting shown, with either empty needle ("Air") or needle filled with low viscosity injectate and held into lamp oil bath. **C-D**. With the 112.5 μm diameter needle attached, low viscosity injections were performed on "medium" PWM setting for 30 ms (**C**) or 60 ms (**D**). All micrograph linear scales show mm units.

Reproducibility was assessed with repeat measurements (Fig 6A) for precision and error estimation (Fig 6B). Median relative standard deviations (RSDs) for low-viscosity injectate were typically 14–16%, with maximum error values around 18–22%. Linear fits (after excluding unstable bins: 24 ms at low-viscosity/high-pressure and the shortest pulse duration in each condition in Fig 6B) confirmed strong correlations between pulse duration and volume, with $R^2$ values of 0.76 (high pressure), 0.94 (medium pressure), and 0.91 (low pressure). These results indicate that the absolute volume range and usable pulse duration depend on pressure, and the device achieves predictable scaling of dispensed volume across conditions (after unstable extremes are identified and avoided).

For high-viscosity injectate (142 ± 5 cP; see S1 File for fluid viscosity measurements), reproducible ejections were only possible at the maximum pressure setting, as expected given the resistance to flow. Surprisingly, the correlation between pulse duration and dispensed volume was even stronger than for water ($R^2 \approx 0.98$), with better reproducibility (median RSD ~ 9%, maximum error ~14%). However, the smallest stable volume achievable with high viscosity injectate was larger (~70 nL) than the best-case low-viscosity volumes (~22 nL). This suggests that while viscous fluids dampen variability in flow, they also impose a higher floor on injection volumes, since higher minimum pulse duration is required to overcome surface tension and capillary forces at the needle tip.

Taken together, these results show that the injector is capable of producing highly reproducible nanoliter-scale injections across a wide operational window. Pressure and viscosity jointly determine both the lower and upper limits of usable pulse durations. High pressure and low viscosity permit the smallest achievable injections but introduce instability at longer pulses, while high viscosity enforces a higher minimum injection volume but yields exceptionally linear and reproducible scaling once ejection is underway. These trade-offs mirror practical biological scenarios, where the choice of injectate and required dose will dictate optimal operating windows.

 

Injectate: Low Viscosity (~1 cP) — High Viscosity (~142 cP)

Pressure: High / Medium / Low / High

**Fig 6. Quantitative performance of the open-source microinjector. A**. Categorical jitter plots showing all individual droplet-volume measurements across pulse durations for each condition: low-viscosity injectate (water, ~1 cP) at high, medium, and low pressure, and high-viscosity injectate (maple syrup, ~200 cP) at high pressure. Boxplots denote interquartile range and median; black dots represent individual injections. **B**. Continuous plots showing mean ± SE (black circles) for each pulse duration with relative standard deviation (blue circles) and maximum error (red squares) displayed on a log-scaled secondary y-axis. Linear regressions yielded $R^2 = 0.76$ (high-pressure low-viscosity), 0.94 (medium), 0.91 (low), and 0.98 (high-pressure high-viscosity). Median RSDs were ~14–16% for low-viscosity and ~9% for high-viscosity conditions. The smallest reproducible droplets measured ≈22–25 nL for low-viscosity/high-pressure and ≈70 nL for high-viscosity/high-pressure injections.

## Discussion

We describe a low-cost, open-source microinjector that pairs a compact, PWM-driven diaphragm pump with bidirectional valve control and footswitch actuation to enable rapid, two-handed microinjection without external compressed gas, custom machining, soldering, or 3D printing. In bench testing the device reliably dispensed nanoliter-scale volumes, with median relative errors near ~15% for low-viscosity fluids and improved precision for higher viscosity fluids. The optional pulse controller (Fig 7) provided millisecond-to-second timing with excellent repeatability and was essential for disentangling device performance from human timing variability (Fig 8).

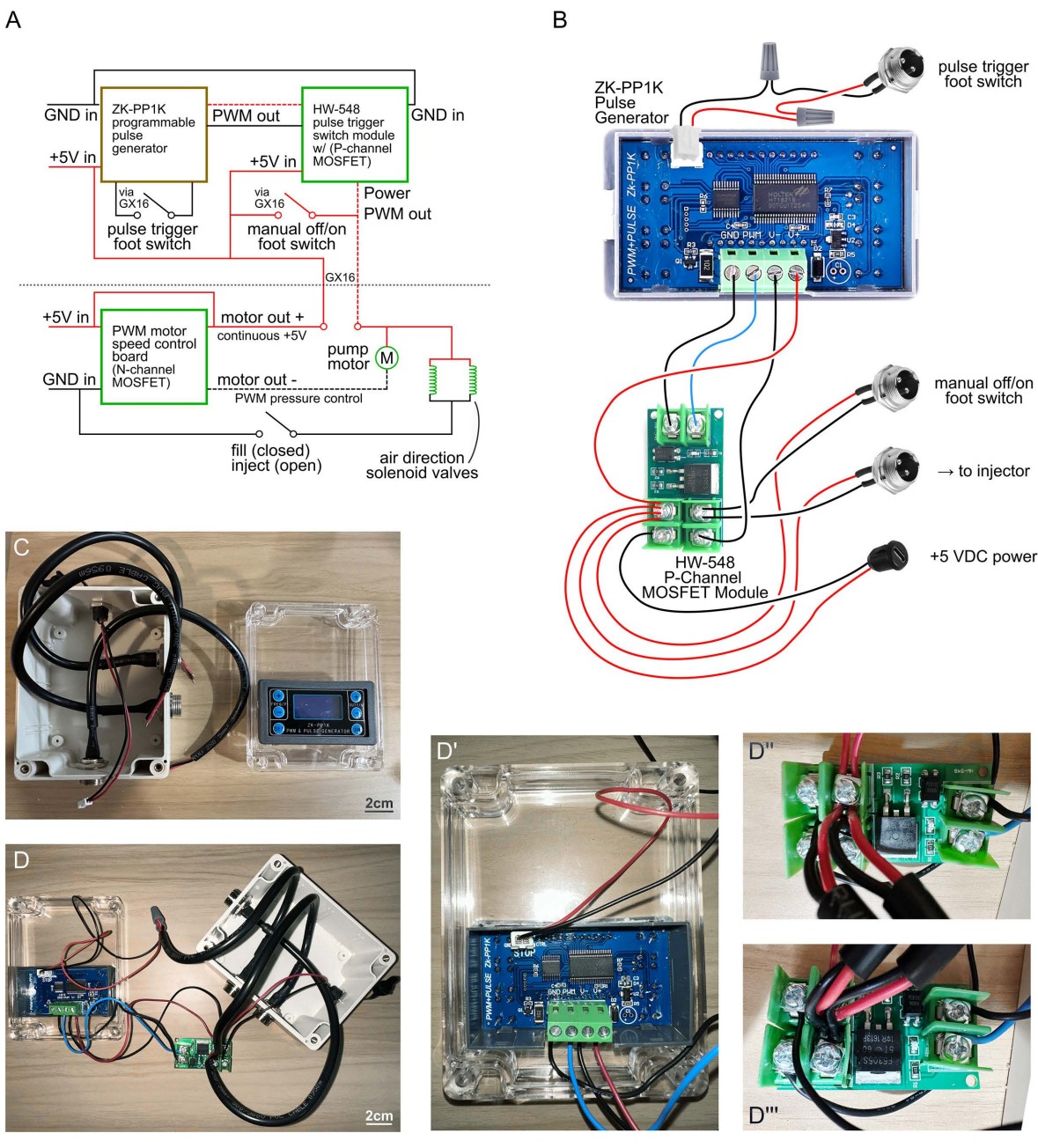

**Fig 7. Pulse Controller Configuration. A**. Block diagram of electronic control modules in the main microinjector project (below dashed line) and in the separate (optional) pulse controller (above dashed line), which facilitates tightly-timed pump operation of the microinjector to deliver controlled, reproducible injections. The pulse controller is powered together with the microinjector, at 5V via USB-C connection. Low-cost, off-the-shelf ZK-PP1K pulse generator and HW-548 P-channel MOSFET switch boards comprise the major internal components of the pulse controller. 5V supply is fed from the left footswitch pin of the microinjector via GX16 patch cable, which is high-side switched at the HW-548 MOSFET and returned to the microinjector, where it is low-side switched for motor speed modulation via PWM. The high-side switch results in timing control gated by ZK-PP1K, and the low-side switch allows controllable pressure via user-selectable dial. Two footswitches can be connected to the pulse controller to trigger 1. pulse initiation by ZK-PP1K and 2. for manual bypass operation (i.e., for aspirations or foot-controlled timing). **B**. Wiring diagram of the pulse controller, for user assembly. Detailed parts list is available in S3 Table. **C**. Beginning of assembly. In the enclosure base, drill three 5/8" spade holes for GX16 ports and one 5/8" regular holes for USB-C power port. Install three GX16 male panel mount plugs and a USB-C female power port into the enclosure base. Mark the rectangular outline of the ZK-PP1K panel mount on the enclosure lid, drill periodic (corners and every 10 mm or so) sentinel holes with the 1/8" regular bit, then connect the sentinel holes with the saw bit on the rotary tool (i.e., Dremel brand). After removing the rear panel of the ZK-PP1K module, install the module into the window you created. **D**. Make all connections shown in (**B**). **D'**. Expanded view of rear of ZK-PP1K module. **D"-D'''**. Expanded two-angle view of connections on HW-548 board.

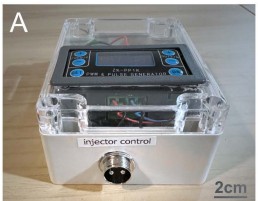
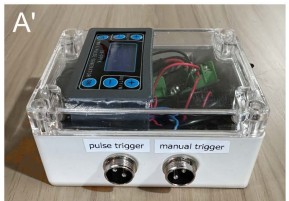

**B**

USB-C
power splitter

5V 3A USB-C
power supply

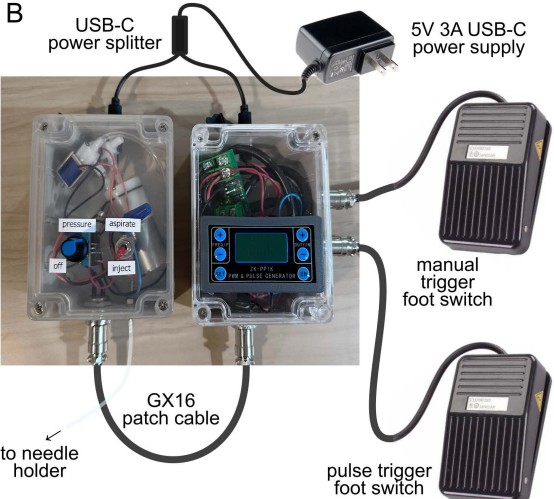

pressure   aspirate

off        inject

GX16
patch cable

to needle
holder

manual
trigger
foot switch

pulse trigger
foot switch

**C**

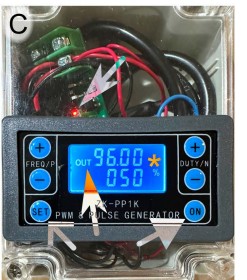

OUT 96.00 ✱
0.50 %

pulse controller starts
with pulse output on

➤ "OUT" and HW-548 LED illuminated

➤ press "ON" or depress pulse trigger
footswitch to turn off

two pulse modes: continuous PWM
or triggered pulse mode

✱ continuous PWM mode shows "%"

→ to switch to triggered pulse mode,
long hold (>6s) "SET" button

**D**

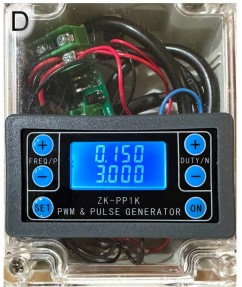

0.150
3.000

**D'**

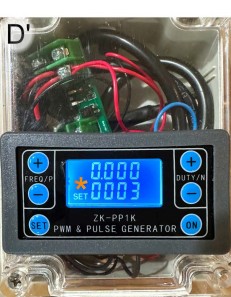

0.000
SET 0003

triggered pulse mode - 4 settings on 2 screens

→ short hold (>6s) "SET" button switches between screens

✱ "SET" screen: sec trigger delay (usually 0) / pulse number (usually 1)

blank screen: sec pulse duration (user set) / sec pulse interval (usually 1-3)

**Fig 8. Pulse Controller Operation. A-A'.** Final assembly of the pulse controller, with exterior GX16 jacks shown. The "injector control" port is connected via GX16 patch cable to the microinjector footswitch port. "Trigger" ports (**A"**) are connected to footswitches, one each for 1. triggering the controlled pulses and 2. manual start/stop. **B**. External connections of the main microinjector and pulse controller, which should be made prior to power-on. **C**. Startup operation of the microinjector and pulse controller. When powered, the pulse controller begins with pulse output denoted by "OUT." HW-548

board LED will illuminate in synchrony with the pulses as they are delivered. To terminate pulse output, press "ON" or the pulse trigger footswitch. When off ("OUT" not illuminated), the pulse output can be restarted by depressing the pulse trigger footswitch. On first start, the ZK-PP1K module may enter PWM Mode denoted by "%", with continuous uninterrupted pulse output. For microinjector use, we desire triggered pulse mode, which can be entered by long pressing the "SET" button (>6 seconds), and is appreciated when "%" is no longer illuminated. **D**. In triggered pulse mode, four settings are managed on two LCD screens in ZK-PP1K that are switched with short pressing the "SET" button (>2 seconds): The "SET"-illuminated screen shows trigger delay (s) on top and pulse number (#), while the non-"SET" screen shows pulse duration (s) on top and pulse interval (s) on bottom. The +/- buttons increment or decrement the values. Typically, trigger delay is 0, pulse number is 0, and pulse duration is adjusted to desired injection duration.

## Relationship to prior open hardware

Several open devices address micro- and pico-scale delivery but with different trade-offs (Table 1). Pressure-puff systems like Openspritzer [9] require external regulation and typically provide dispense-only operation; Arduino syringe-pump designs [10,14] excel at rate control but are slower for spot injections and usually lack an integrated aspirate mode; organism-specific tools based on microfluidics or PDMS [12,13] achieve elegant positioning but entail specialized fabrication; and the 3D-printed autoinjector targets a milliliter-scale clinical niche [15]. In contrast, our system emphasizes standalone operation, dual-mode (aspirate/inject) flow control, hands-free triggering, fast assembly with commodity parts, and benefits labs that need sub-microliter injections for *in vivo* manipulation but lack access to compressed air or fabrication facilities.

## Practical operation: pressure × pulse × viscosity

Our measurements clarify how pressure and viscosity define the usable pulse window. At high pressure, very short pulses (≤10 ms) produced the smallest stable droplets (≈22–25 nL). At lower pressures, the valid pulse window shifted to longer durations, which paradoxically increased the minimum stable volume (e.g., ~45–50 nL after excluding unstable first bins). Thus, if the experimental goal is minimal dose, higher pressure with brief (millisecond-scale, requiring electronics-based timing), timed pulses is advantageous. If robust stability over a wider duration range is preferred, moderate pressure can be more forgiving (especially for human manual actuating) albeit with larger minimum volumes.

Fluid viscosity further shapes performance. With low-viscosity injectate (~1 cP), median RSDs clustered around 14–16% across pressure settings. With high-viscosity maple syrup (~142 cP at room temperature [16,17]), reliable ejection occurred only at maximum pressure, but once flow initiated the volume-versus-time relationship was extremely linear ($R^2 \approx 0.98$) and precision improved (median RSD ~9%). However, the smallest achievable volume at high viscosity was larger (~70 nL) than for water, given the higher surface tension and capillary force thresholds at the needle tip before stable flow is established. These findings are consistent with principles of viscous damping [20] that reduce jet breakup to yield more linear and reproducible volume scaling with increasing viscosity. In practice, investigators can emulate planned injectates prior to live experimentation with sucrose or glycerol solutions, and select pressure/pulse settings that balance minimal dose against stability.

## Reproducibility, calibration, and measurement considerations

Across low-viscosity conditions, linear fits of dispensed volume versus pulse duration remained strong after excluding known unstable settings ($R^2 = 0.76$–0.94). These relationships support employing simple, per-experiment calibration curves that map pulse duration to expected volume at a given pressure and needle geometry. We measured volumes from droplets formed in lamp oil using macro imaging and spherical approximation; this method is convenient, inexpensive, and repeatable, but it can over- or under-estimate volume if droplets deviate from sphericity or fragment at impact. For this approach in practice, users should adopt consistent imaging geometry, include a ruler in frame, and avoid pulse duration/ pressure pairs prone to instability.

## Fluid-dynamics explanations for boundaries of viable dispense parameters

At the needle tip, flow initiation must overcome the capillary (Laplace) pressure associated with the curved meniscus, which scales as $\Delta P_{needle} \approx 2\gamma/R$, where $\gamma$ is surface/interfacial tension and (R) is the meniscus radius of curvature [21]. Consequently, at low pressure settings and/or very short pulses, a pressure transient may be insufficient to initiate measurable ejection. Once flow begins, pressure-driven transport through a narrow needle is typically well approximated by viscous (Poiseuille) scaling, in which volumetric flow rate increases linearly with pressure and decreases with viscosity and needle length, with a quartic dependence on needle inner radius ($Q \propto r^4 \cdot \Delta P/(\eta L)$) [22]. Separately, at higher pressure levels droplet formation can transition toward jetting/filamenting, where a liquid column forms and then breaks up downstream via capillary (Rayleigh–Plateau) instability and may produce satellite droplets [23]. Although these principles influence the performance of all micro-injector devices, the qualitative and measured behavior of this open source microinjector abundantly exemplifies them.

## Applications

Our design of the microinjector was motivated by *in vivo* electroporation in the mammalian forebrain, particularly *in utero* [3,4,24,25] and postnatal contexts [6,26]. In these settings, rapid, precise injections of plasmids or reagents into small compartments (e.g., lateral ventricles or ventricular zones) are crucial to minimize exposure time, maintain embryo viability, and reduce diffusion artifacts. Although designed with neuroscience applications in mind, we first applied this injector toward lipophilic dye labeling of cellular cohorts in early mouse embryos *ex vivo* [27,28], a forerunner to *in toto* imaging and massive computational reconstruction [29] that ultimately obviated sparse labeling. Moreover, we have used this same injector to deliver reagents by intra-oviductal electroporation for *in situ* genome editing in mice, leveraging iGONAD-style strategies [7,30] to bypass *ex vivo* zygote handling and embryo transfer. In our hands, the injector allows exceptional control of injected volumes into the oviduct lumen, freeing both hands for manual manipulation of miniscule rodent oviducts and injection needle. Because the device is intended as a general-purpose microinjection platform, biological outcomes will depend on the specific experimental system and are therefore best evaluated within the context of each application.

Other potential applications for this microinjector (beyond electroporation) are numerous. The ability to aspirate and dispense small volumes under footswitch control opens opportunities in viral transduction, by delivering low-volume viral vectors (AAV, lentivirus) into brain, retina, muscle, or peripheral organs with minimal spread and leakage. In developmental model organisms (zebrafish, *Xenopus*, *Drosophila*, *C. elegans*), sub-microliter injections of mRNA, CRISPR/Cas complexes, morpholinos, or tracers can be executed with greater dexterity and accuracy compared to manual mouth pipetting or slow syringe pumps. Localized drug or gene delivery to specific regions within a spheroid, organoid, or other 3D culture is possible. In physiology or pharmacology, microinjection of microdoses into tissue slices or muscle bundles may allow highly localized testing [31,32]. Finally, because the device is compact and self-powered (no bulky compressed gas needed), it is suitable for field or low-resource laboratory environments, such as developmental biology in amphibian field stations, or educational settings where standard microinjection infrastructure is unavailable.

When operated in continuous PWM mode using the optional pulse controller, the pump can deliver a low, steady compensation or ballast pressure—analogous to "hold" pressure in commercial microinjectors, rendering utility in delicate tasks such as ICSI, blastocyst injection, adherent cell microinjection, and RNAi delivery.

## Advantages for *in vivo* manipulation

Hands-free actuation enables true two-handed work at the tissue while maintaining rapid aspirate→inject transitions, which is valuable in embryo and small-animal procedures (e.g., in utero electroporation, stereotaxic viral delivery, retinal or cardiac microinjection). The aspirate mode permits quick loading and reagent changes without removing the needle, facilitating sequential injection of multiple reagents quickly within a single session. The standalone, < USD $130 bill of materials lowers barriers for teaching labs and resource-limited settings, potentially broadening access to *in vivo* genetic manipulation techniques.

### Shear forces, acoustic noise, and contamination considerations

The injector operates using a low-power diaphragm air pump to generate pressure pulses, which are transmitted to the needle via tubing with modest capacitance. As such, shear stresses within the capillary are expected to be relatively low compared with syringe-driven injection systems that rely on rapid piston displacement. In practice, this operating mode is well suited for biological suspensions such as plasmid solutions, viral vectors, or cell-compatible dyes, which can be sensitive to excessive shear. The miniature diaphragm pump produces a modest audible hum during operation but is substantially quieter than compressed-air regulators or large laboratory pumps typically used with commercial pressure injectors.

As with any microinjection system, care should be taken to minimize cross-contamination between injectates. In many experimental workflows—such as viral delivery or electroporation—glass needles are routinely replaced between animals or preparations, which naturally minimizes cross-contamination risk. For maximal mitigation of cross-contamination, we recommend our custom home-built needle holder (lower panel in Fig 2B), for which metal components are autoclavable, and plastic or rubber parts cheaply replaceable (including tubing).

### Limitations

First, while our pump/valve architecture is compact and accessible, absolute volume accuracy is influenced by tubing compliance, backpressure at the needle tip, and fluid properties; laboratories should expect to establish local calibration curves on a per-experiment basis because delivered volume depends on needle tip geometry, pressure setting, and injectate viscosity. Second, our approach does not provide closed-loop pressure or flow feedback; thus, edge cases (very high backpressure tissues or extremely short pulses below 1–10 ms) may be sensitive to variability in injection response. In measuring injections, our volume assay assumes near-spherical droplets in oil; for highly wetting fluids or very large ejections, shape assumptions break down. On a practical note, sterility, decontamination, and electrode positioning (for electroporation) remain user-dependent; the system does not replace good surgical practice.

### Future directions

Several extensions are straightforward within the same open-hardware mindset: (i) integrating a miniature pressure sensor into the pump for logging real-time pressure; (ii) microcontroller-based timing in place of the off-the-shelf pulse generator, enabling programmable pulse trains, pressure ramps, or automatic aspirate-then-inject macros; (iii) compact, swappable needle-tip modules (filters, anti-drip valves, or hydrophilic/hydrophobic coatings) to mitigate tip surface-tension effects at the smallest volumes; (iv) optional battery power for field or stereoscope setups without convenient AC mains access; and (v) community-maintained variants and printed accessory brackets.

### Materials, disposal, and sustainability considerations

The injector, which is intended for repeated use, is assembled primarily from durable components including small electromechanical parts, plastic tubing, and standard electronic modules. The only routinely replaced components are glass microinjection needles or capillaries, which are standard laboratory consumables and are discarded following normal laboratory sharps or biohazard disposal procedures depending on the experimental context. Because the device requires no compressed gas cartridges or single-use syringes, routine operation generates relatively little consumable waste. While the materials used in the injector are not biodegradable, the design emphasizes long-term reuse of parts and replacement of inexpensive tubing or needles when necessary, which reduces overall material consumption compared with some commercial microinjection systems.

### Conclusion

This microinjector demonstrates that reliable, nanoliter-scale injections can be achieved with an accessible, fully open design that assembles in two hours and operates without compressed gas. The combination of footswitch control,

dual-mode aspiration/dispense, and optional add-on pulse controller yields practical performance for *in vivo* electroporation, viral transduction, lineage tracing, and related applications. With openly available design, build steps, and validation, we hope to catalyze broad adoption and community customization.

## Supporting information

**S1 Table: Detailed Parts List.**
(ODS)

**S1 Video. Microinjector Setup and Operation.**
(MP4)

**S2 Video. Pulse-Controlled Example Injections.**
(MP4)

**S1 File. Source Data and Code.**
(ZIP)

## Author contributions

**Conceptualization:** Kevin B. Hayes, Martin H. Dominguez.

**Data curation:** Victor H. Dominguez.

**Funding acquisition:** Mark L. Kahn, Martin H. Dominguez.

**Investigation:** Victor H. Dominguez, Maxwell Frankfurter, Martin H. Dominguez.

**Methodology:** Victor H. Dominguez, Kevin B. Hayes, Martin H. Dominguez.

**Resources:** Mark L. Kahn.

**Supervision:** Mark L. Kahn, Martin H. Dominguez.

**Validation:** Victor H. Dominguez, Maxwell Frankfurter.

**Writing – original draft:** Victor H. Dominguez, Martin H. Dominguez.

**Writing – review & editing:** Victor H. Dominguez, Maxwell Frankfurter, Kevin B. Hayes, Martin H. Dominguez.

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
