## [Decision Letter · Decision Letter 0]

30 Jan 2026

PONE-D-25-60637A portable, ultra-low cost, open-source, pedal-controlled microinjector for laboratory usePLOS One

Dear Dr. Dominguez,

Thank you for submitting your manuscript to PLOS ONE. After careful consideration, we feel that it has merit but does not fully meet PLOS ONE’s publication criteria as it currently stands. Therefore, we invite you to submit a revised version of the manuscript that addresses the points raised during the review process.

We look forward to receiving your revised manuscript.

Kind regards,

Yufeng Zhou, PhD

Academic Editor

PLOS One

Journal Requirements:

5. We are unable to open your Supporting Information file generate_categorical_plots_all_data.py and generate_continuous_plots_errors.py. Please kindly revise as necessary and re-upload.

Reviewers' comments:

Reviewer's Responses to Questions

**Comments to the Author**

1. Is the manuscript technically sound, and do the data support the conclusions?

Reviewer #1: Yes

Reviewer #2: Yes

2. Has the statistical analysis been performed appropriately and rigorously? 

Reviewer #1: Yes

Reviewer #2: Yes

3. Have the authors made all data underlying the findings in their manuscript fully available?

Reviewer #1: Yes

Reviewer #2: Yes

4. Is the manuscript presented in an intelligible fashion and written in standard English?

Reviewer #1: Yes

Reviewer #2: Yes

5. Review Comments to the Author

Reviewer #1: This report concerns the manuscript entitled “A portable, ultra-low cost, open-source, pedal-controlled microinjector for laboratory use”.

After reading the manuscript, I believe that this work has very useful information and it should be published after some considerations that I hope will enhance the impact and relevance. The authors provide a very well written manuscript and clear instructions on how to build a cost-effective device.

In the Introduction part, can some details be given about the delivery timescale, and penetration depth of the envisaged injections? Assuming that the system is not agnostic to the fluid type, can a list of fluids with physicochemical properties range be given to educate the future users? E.g., around line 135, Typical injection/aspiration times, at least for water? Depending on materials/fluid - would it work with non-aqueous liquids?

When stating that inspection/checking for leaks, is it meant by optical (eye)inspection for droplets? Or ‘gas’ leaks … how?

Is it possible to say something about the use biodegradable materials? What happens after usage of components, such as the needle that has been in contact with different substrates?

Could the authors discuss some considerations about recycling/sustainability?

Near line 160, Are there references about red cabbage as biological surrogate?

Near line 165, can the authors be more specific about the lamp oil? E.g., manufacturer or cP or other ‘units’?

Line 170, are the large bubbles observed due to air entrainment during droplet impact on the lamp oil? Some videos or images would be very helpful, e.g., around line 185, Visualizations of the injection into the lamp oil?

Line 190, Is it possible to show pictures of how the device is ‘held’ by somebody’s hands.

Ergonomic aspects not needed at this stage, but could be useful to say something about it.

Line 245, Any risks of cross-contamination in between changing needle, or vapor residues in the tubes ?

Line 270, Can some explanation be given, and perhaps backed up by literature about fluid dynamics of such process of droplet/jet ejection instabilities?

Line 290, Is it possible to provide some fluid dynamics considerations/equations that relate the fluidic properties with the pressure differences and observables: perhaps simple Bernoulli does a good job.

Line 325, Is it possible to provide a video showing apsire/inject operation?

How noisy it is to operate?

Line 330, what are the estimated shear forces that the injectate will experience in the inject/aspire process?

Some injectable fluids may be sensitive to shear. How robust are plasmid and other molecules envisaged by the authors?

Line 370, Can a citation be given about agonist/antagonist effects?

Suggestions:

Recent work on small volume (Drug delivery) injection methods has been covered in recent reviews:

1. Schoppink, Jelle, and David Fernandez Rivas. "Jet injectors: Perspectives for small volume delivery with lasers." Advanced drug delivery reviews 182 (2022): 114109.

2. Phatale, Vivek, et al. "Overcoming skin barriers through advanced transdermal drug delivery approaches." Journal of controlled release 351 (2022): 361-380.

Reviewer #2: This work titled "A portable, ultra-low cost, open-source, pedal-controlled microinjector for laboratory use" reports a low-cost, open-source microinjector that pairs a compact, PWM-driven diaphragm pump with bidirectional valve control and footswitch actuation to enable rapid, controllable microinjection. Overall, the work is practical and interesting, demonstrating considerable technical skill from the researchers. However, several issues should be addressed before the manuscript can be considered for publication.

Comment 1: The air pump and solenoid valve are the core components of this injector. It is recommended to provide a more detailed introduction to their working principle and the differences in switching between suction/injection modes.

Comment 2: In lines 34 and 77, the authors mention that the system is capable of "enabling multiplexed injections or rapid switching between reagents within a single experiment." However, the Materials and Methods section and Results section do not describe how this functionality is implemented or demonstrate its use. The author needs to clarify or experimentally verify this feature.

Comment 3: In line 359-364, the authors state that the injector was applied to mouse embryo injections and cite references [7, 21–24]. Yet, none of these references appear to have used the injector described in this work. Suggest the author to supplement relevant supporting materials.

Comment 4: For a study focused on the microinjection device, a key aspect is demonstrating its effectiveness. The authors are encouraged to supplement a video demonstration of the system in operation.

Comment 5: Suggest adding demonstration experiments for practical application scenarios, such as microinjection of certain zebrafish or mouse embryos.

Comment 6: A critical technical concern is how consistent output volume is maintained when needle diameter varies, which can occur when using a pipette puller to prepare glass needles.

6. PLOS authors have the option to publish the peer review history of their article (what does this mean?). If published, this will include your full peer review and any attached files.

Reviewer #1: No

Reviewer #2: No

---

## [Author Response · Author response to Decision Letter 1]

10 Mar 2026

Please see the attached Response to Reviewers document file for point-by-point description of reviewer feedback and our replies.

---

## [Decision Letter · Decision Letter 1]

1 Apr 2026

A portable, ultra-low cost, open-source, pedal-controlled microinjector for laboratory use

PONE-D-25-60637R1

Dear Dr. Dominguez,

We’re pleased to inform you that your manuscript has been judged scientifically suitable for publication and will be formally accepted for publication once it meets all outstanding technical requirements.

Kind regards,

Yufeng Zhou, PhD

Academic Editor

PLOS One

---

## [Editor Report · Acceptance letter]

PONE-D-25-60637R1

PLOS One

Dear Dr. Dominguez,

I'm pleased to inform you that your manuscript has been deemed suitable for publication in PLOS One. Congratulations! Your manuscript is now being handed over to our production team.

Kind regards,

on behalf of

Dr. Yufeng Zhou

Academic Editor

PLOS One